# Daily Aspirin Reduced the Incidence of Hepatocellular Carcinoma and Overall Mortality in Patients with Cirrhosis

**DOI:** 10.3390/cancers15112946

**Published:** 2023-05-27

**Authors:** Chern-Horng Lee, Chiu-Yi Hsu, Tzung-Hai Yen, Tsung-Han Wu, Ming-Chin Yu, Sen-Yung Hsieh

**Affiliations:** 1Division of General Medicine and Geriatrics, Chang Gung Memorial Hospital, Linkou Branch, Taoyuan 333, Taiwan; lee4570@cgmh.org.tw; 2Center for Big Data Analytics and Statistics, Chang Gung Memorial Hospital, Linkou, Taoyuan 333, Taiwan; joy960111@gmail.com; 3Department of Nephrology, Chang Gung Memorial Hospital, Linkou Branch, Taoyuan 333, Taiwan; 4Department of General Surgery, Chang Gung Memorial Hospital, Linkou, Taoyuan 333, Taiwan; 5Department of Gastroenterology and Hepatology, Chang Gung Memorial Hospital, Linkou Branch, Taoyuan 333, Taiwan; 6College of Medicine, Chang Gung University, Taoyuan 333, Taiwan

**Keywords:** aspirin, antiplatelet agents, cirrhosis, liver cancer, hepatocellular carcinoma, chemoprevention

## Abstract

**Simple Summary:**

The impacts of daily low-dose aspirin on hepatoma occurrence and gastrointestinal bleeding incidence in cirrhotic patients remain unknown. This retrospective study enrolled 66,984 cirrhotic patients with laboratory data, the largest cirrhotic cohort. Aspirin users, but not those of other antiplatelet agents, showed a dose-dependent reduction in hepatoma incidence and overall mortality. Daily aspirin did not increase the gastrointestinal bleeding incidence rate in cirrhotic patients.

**Abstract:**

Background: Cirrhosis is the primary risk factor for hepatocellular carcinoma (HCC) and gastrointestinal bleeding (GI). We aimed to assess the efficacy and safety of daily aspirin on HCC occurrence, overall survival, and GI bleeding in cirrhotic patients. Methods: A total of 35,898 eligible cases were enrolled for analyses from an initial 40,603 cirrhotic patients without tumor history. Patients continuously treated with aspirin for at least 84 days were in the therapy group, whereas those without treatment were controls. A 1:2 propensity score matching by age, sex, comorbidities, drugs, and significant clinical laboratory tests with covariate assessment was used. Results: Multivariable regression analyses revealed that daily aspirin use was independently associated with a reduced risk of HCC (three-year HR 0.57; 95% CI 0.37–0.87; *p* = 0.0091; five-year HR 0.63, 95% CI 0.45–0.88; *p* = 0.0072) inversely correlated with the treatment duration [3–12 months: HR 0.88 (95% CI 0.58–1.34); 12–36 months: HR 0.56 (0.31–0.99); and ≥ 36 months: HR 0.37 (0.18–0.76)]. Overall mortality rates were significantly lower among aspirin users compared with untreated controls [three-year HR 0.43 (0.33–0.57); five-year HR 0.51 (0.42–0.63)]. Consistent results were obtained when the laboratory data were included in the propensity score for matching. Conclusions: Long-term aspirin use significantly reduced the incidence of HCC and overall mortality without increasing gastrointestinal bleeding in cirrhotic patients.

## 1. Introduction

Cirrhosis is the end stage of progressive liver fibrosis, resulting in distorted liver architecture, impaired liver function, and portal hypertension. Even in the post-viral hepatitis era, the prevalence of cirrhosis remains high, with high morbidity and mortality [1]. The primary challenges associated with the medical care of cirrhotic patients are an increased incidence of hepatocellular carcinoma (HCC) and portal-hypertension-associated complications, particularly gastrointestinal (GI) bleeding; cirrhosis is one of the leading causes of death globally [1].

HCC is the seventh-most prevalent cancer type and the second-most common cause of cancer mortality worldwide [2]. Approximately 90% of HCC develops in cirrhotic livers. Cirrhosis is, indeed, the most critical risk factor for HCC [3]. The 5-year cumulative rates of HCC are 17–30%, 10–15%, 8%, and 5.3% in patients with hepatitis C virus (HCV)-, hepatitis B virus (HBV)-, alcohol-, and nonalcohol fatty liver disease (NAFLD)-related cirrhosis, respectively [3,4]. Despite the high efficacy of HBV vaccination of newborns and the high curative rate of HCV infection with direct antiviral agents, the rapid rise in the prevalence of metabolic syndrome, obesity, and type II diabetes has become the primary cause of cirrhosis and HCC globally. Moreover, HCC is the leading cause of mortality in compensated cirrhosis [1]. Therefore, treatments to reduce HCC incidence in cirrhotic patients represent a critical unmet need.

In high-risk populations such as those with cirrhosis, chemoprevention (using natural or synthetic compounds to inhibit cancer development) is appealing [5,6]. Platelets have been reported to promote tumor growth and metastasis, including HCC [7,8]. The involvement of platelets goes beyond the direct tumorigenic effects on tumor cells, as they are known to play a role in pro-fibrogenic signaling and induction of proinflammatory cytokines in the microenvironment to foster tumor formation and progression [9,10]. Platelets can be released by multiple factors, including TXA2, ADP, angiogenic factors (VEGF, FGF, PDGF) and growth factors (IGF-I, TGF-β1, SDF-1), and direct interaction with leucocytes and endothelial cells, subsequently promoting cancer cell proliferation, angiogenesis, and metastasis [11]. Clinically, thrombocytosis is associated with high malignancy, metastasis, and poor prognosis [7,12]. Therefore, platelet activation inhibitors are promising candidates for the chemoprevention of HCC [13]. Of the platelet activation inhibitors, aspirin is of particular interest because of its potential suppression of cellular tumor transformation [14], immune-mediated chronic hepatitis [15], and liver fibrosis [16] through diverse mechanisms, such as inhibiting platelet activation and proinflammatory cyclooxygenase-2 [17,18]. Recently, several nationwide cohort studies revealed that daily aspirin use reduced the incidence of HCC in patients with chronic hepatitis B or C [19,20,21] and health care professionals [21]. However, these studies are restricted by the limited number of cirrhotic cases and the lack of comprehensive laboratory data related to the occurrence of HCC. Aspirin therapy was associated with only a small increased risk of gastrointestinal bleeding in patients with hepatitis B or C virus, alcohol-related liver disease, or nonalcoholic steatohepatitis [11]. Given that cirrhosis poses the primary risk for HCC and is highly susceptible to GI bleeding, it is imperative to examine the efficacy and safety of daily low-dose aspirin and other antiplatelet agents (APAs) in patients with cirrhosis.

The Chang Gung Research Database (CGRD), a regularly updated and well-validated tool, offered comprehensive diagnostic, laboratory, medication, and outpatient and hospitalization information of patients followed in the long term at six leading hospitals in different regions of Taiwan. We used propensity score matching (1:2) to randomly select patients with or without aspirin or APA treatment (≥84 days) to evaluate their efficacy in HCC prevention and the risk of GI bleeding in cirrhotic patients.

## 2. Materials and Methods

### 2.1. Data Sources

The data used in this study were retrieved from the CGRD during the period from 1 January 2003 to 31 December 2017. The CGRD included all patients’ information from six Chang Gung Memorial Hospitals in different regions of Taiwan, encompassing approximately one-fourth of the Taiwan population (5.8 million citizens). This database contains primary patient demographic data, clinic visits, hospitalization dates, diagnoses, clinical studies, prescriptions, and all laboratory data. A diagnosis of cirrhosis was based on at least three clinic visits or hospitalization with imaging studies, symptoms and signs of portal hypertension, and some histological evidence of liver cirrhosis. Comorbidities were retrieved using the ICD-9 (from 1997 to 2015) and ICD-10 (since 2016) codes (shown in Appendix A online) and diagnosed at least three times in outpatient clinics or once during hospitalization. Drugs were identified according to WHO anatomical therapeutic chemical (ATC) codes (shown in Appendix A online). The observation period was from the index date to the end of 2017, death, or the development of HCC, whichever occurred first. The CGRD database was linked to the National Patient Registry, Cancer Registry, and Prescribed Drug Registry databases. The Institute Review Board of the Chang Gung Medical Foundation approved the study and waived informed consent.

### 2.2. Study Population

As shown in (Figure 1), 66,984 cirrhotic patients without tumors were enrolled. After exclusion of those with previous malignancy, age under 20 years, follow-up duration <30 days, HCC diagnosed < 30 days, and mortality < 30 days, 35,898 patients were included in the subsequent analyses. To evaluate the impacts of daily aspirin (75–100 mg/day) or APAs on HCC incidence, we excluded those with daily aspirin or APA treatment < 84 days, resulting in 32,644 and 33,959 cirrhotic patients, respectively. Treatment defined as the duration of at least 84 days was based on the Taiwan National Health Insurance Program reimbursement policy, in which 28 days, instead of 30 days, were counted as one month. Finally, 1135 and 2270 patients with or without daily aspirin were identified by using a 1:2 propensity score to match with standardized mean difference <0.1 for age, sex, cardiovascular diseases, cerebral vascular diseases, diabetes mellitus, and medication with metformin or statins (Study A, Figure 1). We further added the laboratory data related to HCC occurrence, including serum alanine aminotransferase (hepatitis activity) [22], platelet counts (liver fibrosis degree) [23], and total bilirubin and prothrombin time (cirrhosis severity) to the matching score, resulting in 769 and 1538 cirrhotic patients with or without daily aspirin (Study B). The sample criteria for propensity score matching were also applied to selecting patients with or without daily antiplatelet agents other than aspirin (APAs; Figure 1, Study C and D).

### 2.3. Main Outcome

The two primary outcomes were HCC incidence and mortality, which were ascertained from the Cancer and Cause of Death registries. The primary goals of the end of the study were HCC diagnosis or death after use or nonuse of aspirin or APAs. If another disease occurred around the index date, it was considered baseline comorbidity, referred to as ICD-9-CM code or ICD-10-CM code (shown in Appendix A online). The follow-up period was defined as the duration between the index date and the date of occurrence of HCC incidence or a competing event, whichever occurred first.

### 2.4. Statistical Analysis

The Cox proportional hazard model was used to compare the relative risk of HCC and GI bleeding between cirrhotic patients with or without antiplatelet agent use. The crude, adjusted hazard ratios and 95% confidence intervals (95% CIs) were estimated using a Cox proportional hazard regression model without adjusting gender, age, and confounding factors. We also assessed the effect of antiplatelet agents on the characteristics of different strata of cirrhotic patients. The cumulative incidence of HCC was calculated using Gray’s method and examined for the differences in the two trends using the log-rank test. The Kaplan–Meier method was used to analyze mortality of all causes as a competing risk of HCC incidence.

Using a propensity score approach, the two primary pre-specified analyses focused on the correlation between aspirin use and the risk associated with HCC incidence and mortality. The multivariable-adjusted model included pre-specified prognostic covariates. All analyses were performed using the SAS statistical package, version 9.4 (SAS Institute, Cary, NC, USA). The significance level was set at 2-tail, with a *p*-value < 0.05.

## 3. Results

### 3.1. Patients and Study Design

Of the 40,603 cirrhotic patients without a history of liver tumors, 35,898 cases, including 1260 and 2575 cases treated with daily aspirin ≥ 84 days or APAs ≥ 84, respectively, met the inclusion criteria and were subjected to the following analyses (Figure 1). We used 1:2 PSM for compatible gender, age, underlying diseases related to aspirin or APAs (including cardiovascular and cerebral ischemic diseases), comorbidities possibly contributing to HCC development (diabetes mellitus and fatty liver diseases), and medications related to the use of aspirin or APAs. A total of 1135 and 2270 randomly matched cirrhotic patients with daily aspirin for at least 84 days or without aspirin use were included (Figure 1, study A). Clinical characteristics are shown in Appendix A. The mean patient age at the start of aspirin use was 61.2 years, and the median treatment duration was 524 days (IQR, 201–1240 days). Considering that hepatitis activity and liver fibrosis are risk factors for HCC, we added serum alanine aminotransferase (ALT), serum bilirubin levels, platelet counts, elongation of prothrombin time, and the cirrhosis etiologies of hepatitis B virus (HBsAg) and antibodies against hepatitis C virus to the original propensity score matching (shown in Appendix A). A total of 769 and 1538 randomly matched cases with daily aspirin ≥ 84 days and without aspirin use were selected (Figure 1, study B). Simultaneously, the impacts of APAs on the incidence of HCC were evaluated using the same two propensity score systems as those for the aspirin study. Accordingly, 1491 vs. 2982 cases (shown in Appendix A; Figure 1, study C) and 1018 vs. 2036 cases (shown in Appendix A; Figure 1, study D) without or with the inclusion of the laboratory data for matching, respectively, were identified for subsequent analyses.

### 3.2. Daily Aspirin and Non-Aspirin Antiplatelet Agents Reduce HCC Incidence

We first compared the HCC incidence rates in patients with vs. without daily aspirin (study A) during a five-year follow-up. The 3- and 5-year incidences of HCC in aspirin-treated patients were 4.14% and 7.22%, respectively, whereas those in the untreated group were 7.97% and 10.93%, respectively (shown in Appendix A, upper panel). After adjusting for competing mortality risk, the cumulative incidence of HCC in the aspirin-treated group was significantly lower than that in the untreated group at five years (*p* = 0.0004 by Gray’s test; adjusted HR 0.63; Figure 2A).

On the other hand, the HCC incidence rate was lower in the APA treatment group than in the untreated group at 3 and 5 years (3 year incidence: 6.24% vs. 8.89%; 5 year incidence: 9.26% vs. 11.94%; shown in Appendix A lower panel; *p* = 0.0062 by Gray’s test). However, the difference was not statistically significant when mortality and other confounding factors were accounted for (adjusted HR = 0.89; *p* = 0.156; shown in Appendix A).

### 3.3. Daily Aspirin Is an Independent Protective Factor of HCC in Cirrhotic Patients

We used Cox regression modeling to explore aspirin and APAs’ efficacy further to reduce the incidence of HCC (Table 1). There was a significant reduction of HCC incidence at 3- and 5-year follow-ups in the aspirin-treated group compared with the untreated group (3-year HR 0.57; 95% CI 0.37–0.87; *p* = 0.009; 5-year HR 0.63; 95% CI 0.45–0.88; *p* = 0.007). However, daily APA did not significantly reduce HCC incidence in cirrhotic patients at 3- (HR 0.82; 95% CI 0.59–1.14; *p* = 0.242) or 5-year follow-ups (HR 0.81; 95% CI 0.61–1.08; *p* = 0.151).

Consistently, when the laboratory data related to the occurrence of HCC were added to PSM, daily aspirin use, but not APA, reduced the incidence of HCC at 3- and 5-year follow-ups (3-year HR 0.47; 95%CI 0.32–0.76; *p* = 0.001; 5-year HR 0.62; 95% CI 0.45–0.88; *p* = 0.007; shown in Appendix A).

### 3.4. Daily Aspirin Is Associated with a Higher Overall Survival Rate

The overall survival probability within five years of follow-up was higher in aspirin users than in untreated patients (adjusted HR 1.96; 95% CI 1.60–2.40; *p* < 0.0001; Figure 2B).

Notably, the overall survival probability within five years of follow-up was higher in APA users than in untreated patients (adjusted HR 1.38; 95% CI 1.18–1.61; *p* < 0.0001; shown in Appendix A).

### 3.5. Daily Aspirin Reduces HCC Incidences by Multivariable Stratified Analysis

Multivariable stratified analysis was used to verify the association of daily aspirin with reduced HCC risk. The results indicated that all the multivariable factors demonstrated an HR of less than 1.0 (Figure 3).

### 3.6. Daily Aspirin Duration Is Inversely Correlated with HCC Incidence

If aspirin can prevent HCC occurrence in high-risk patients with cirrhosis, the effect should show a dose/duration dependence. As shown in Table 2 (upper panel), the risk of HCC was not significantly reduced (3 months to 1 year: adjusted HR 0.88, 95% CI 1.34–0.55) until the duration of daily aspirin use was longer than one year (1 to 3 years, adjusted HR 0.56; 95% CI 0.31–0.996; *p* = 0.049; and >3 years, adjusted HR 0.37; 95% CI 0.18–0.76; *p* = 0.007). Moreover, it appeared that the longer duration of daily aspirin use was associated with a lower risk of HCC (adjusted HR 0.88, 0.56, and 0.37 at 1 y, 3 y, and 5 y, respectively). Similar findings were observed when the laboratory data were added to the original PSM variables (Table 2, lower panel).

### 3.7. Gastrointestinal Bleeding Risk Was Not Increased in Daily Aspirin Users

Cirrhosis tends to cause gastrointestinal (GI) bleeding because of associated portal hypertension, thrombocytopenia, and impaired coagulopathy. Knowing whether daily aspirin increases the risk of GI bleeding in patients with cirrhosis is imperative. Taking into account mortality as a competing risk, Cox regression modeling revealed that at 3- and 5-year follow-up, the incidence of GI bleeding was not increased in daily aspirin users compared to untreated cirrhotic patients without a previous history of GI bleeding (3-year: adjusted HR 0.66; 95% CI 0.49–0.90; 5-year: adjusted HR 0.66; 95% CI 0.50–0.86; Table 3, upper panel) or with a previous history of GI bleeding (3-year: adjusted HR 0.51; 95% CI 0.36–0.71; 5-year: adjusted HR 0.51; 95% CI 0.36–0.72; Table 3, lower panel). Notably, the risk of GI bleeding was also not increased in APA users compared to untreated patients (3-year: adjusted HR 0.71; 95% CI 0.53–0.94; 5-year: adjusted HR 0.66; 95% CI 0.51–0.84; shown in Appendix A).

## 4. Discussion

Cirrhosis is the primary cause of HCC and is associated with susceptibility to comorbidities secondary to portal hypertension, particularly GI bleeding [1]. Aspirin is associated with lower HCC incidence in chronic hepatitis B and C patients [17,19,20,24]. A recent meta-analysis further confirmed these findings [25]. Nevertheless, the efficacy and safety of daily low-dose aspirin in preventing HCC in cirrhotic patients remains to be further clarified. In this study, we took advantage of the CGRD, which contains complete clinical, medication, and laboratory data of patients diagnosed and treated in six leading hospitals in different regions of Taiwan, accounting for approximately one-fifth (>4 million people) of the Taiwanese population. We found that daily low-dose aspirin, but not non-aspirin antiplatelet agents, reduced HCC incidence by 43% and 37% at three and five years of follow-up, respectively, in cirrhotic patients after one year of continuous daily low-dose aspirin. The observed results remained true even after the laboratory data related to HCC occurrence or recurrence were included in the propensity score matching. Moreover, daily aspirin use was not associated with an increased risk of GI bleeding in patients with cirrhosis, regardless of a history of GI bleeding. It is worth noting that long-term APA use was not associated with an increased risk of GI bleeding in cirrhotic patients.

Traditionally, “anti-coagulopathy” is a pathological condition in cirrhotic patients because of thrombocytopenia and reduced coagulating factors generated by the liver. However, evidence suggests a hypercoagulative state, which cannot be detected by routine coagulation tests, to be present in cirrhotic patients and associated with the severity and progression of cirrhosis [26]. Clinical studies have also revealed normal or increased thrombin generation in cirrhotic patients compared with healthy individuals, notwithstanding prolonged prothrombin time [27,28]. In addition, platelet hyperactivity, as evidenced by increased urine excretion of 11-dehydro-thromboxane B2 [29], elevated p-selectin expression on the platelet surface [30], and increased interaction of platelets with von Willebrand factor [31], was noted in cirrhotic patients with thrombocytopenia. Furthermore, low-level endotoxemia resulting from bowel-barrier impairment and bacterial translocation has been proposed as a potential trigger for a clotting cascade in cirrhotic patients [32]. The hypercoagulation state precipitates thrombosis in the microscopic portal system, which might exacerbate portal hypertension and GI bleeding risk. Therefore, we speculate that long-term anti-coagulation and antiplatelet therapies may be helpful to alleviate occult and microscopic portal vein thrombosis and consequently attenuate portal hypertension and associated GI bleeding. A recent study reported an improved prognosis in HCC patients who received antiplatelet therapy [18].

On the other hand, Ma et al. reported that aspirin use was independently associated with a reduced risk of HCC incidence, recurrence, and death, but an increased risk of bleeding with aspirin use was noted [33]. Regina Zi Hwei Tan et al. studied adults with hepatitis B or C virus, alcohol-related liver disease, or nonalcoholic steatohepatitis, but did not focus on cirrhotic patients, and found that administered antiplatelet agent (such as aspirin) use reduced HCC incidence, improved liver-related mortality, and had a small increased risk of gastrointestinal bleeding events [11]. P2Y12 antagonists or reagents directly blocked platelet-derived GPIbα or anti-platelet-therapy-related pathways to block NASH-to-HCC transition [13]. Previous reports show that antiplatelet therapy significantly reduced the risk of HCC after propensity score matching [34]. APA showed a trend of reducing HCC incidence, although less than aspirin efficacy in our cirrhotic patients. Nevertheless, co-administration of proton pump inhibitors or histamine receptor 2 inhibitors with aspirin or APA users might contribute to no increase in GI bleeding risk in cirrhotic patients.

The limitations of this study include its retrospective cohort study design; a follow-up of only 5 years, no NAFLD or NASH stratification analysis, and no analyses on the effects of warfarin, NOAC, or DOAC on HCC incidence, overall mortality, and risk of GI bleeding. Large-scale prospective studies are necessary to validate our observations.

## 5. Conclusions

Daily low-dose aspirin may reduce the incidence of HCC after one year of continuous treatment in cirrhotic patients without an increased risk for GI bleeding. Randomized clinical trials to further validate the benefits and safety of low-dose aspirin as primary chemoprevention of HCC in cirrhotic patients are warranted.

## Figures and Tables

**Figure 1 cancers-15-02946-f001:**
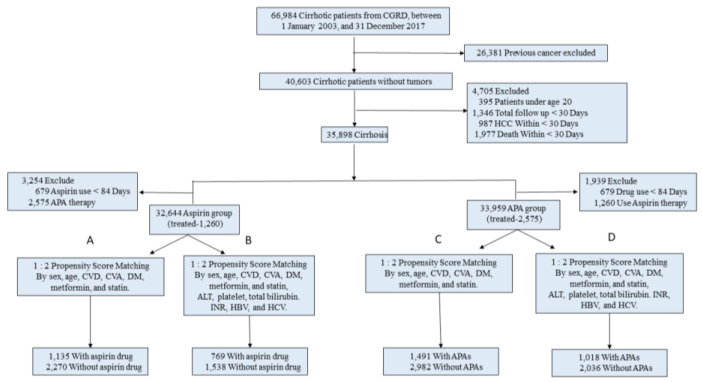
Patient selection and study design. Based on the treatment with aspirin or non-aspirin antiplatelet agents (APAs) and the inclusion or non-inclusion of the critical laboratory data for propensity score matching (PSM), the studies were classified into (**A**–**D**). APA indicates non-aspirin antiplatelet agents; CVD, cardiovascular diseases; CVA, cerebrovascular attacks; ALT, alanine aminotransferase; HBV, positive for serum surface antigen of hepatitis B virus; HCV, positive for serum anti-hepatitis-C-virus antibodies; INR, international normalized ratio of prothrombin time.

**Figure 2 cancers-15-02946-f002:**
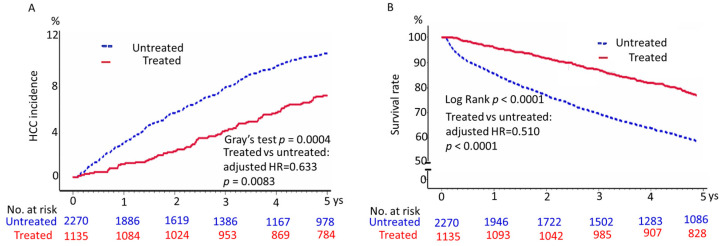
Cumulative incidence of HCC (**A**) and survival (**B**) among aspirin users and nonusers. Aspirin use was defined as a refilled prescription for 84 or more consecutive days of low-dose aspirin after the index date; nonuse was defined as no use. Aspirin use for fewer than 84 consecutive days was excluded. (**A**) *p* value was determined using Gray’s test for equality of the cumulative incidence functions between the two groups after the inverse probability of treatment weighting, accounting for competing risk of death. (**B**) The log-rank test measured the *p* value.

**Figure 3 cancers-15-02946-f003:**
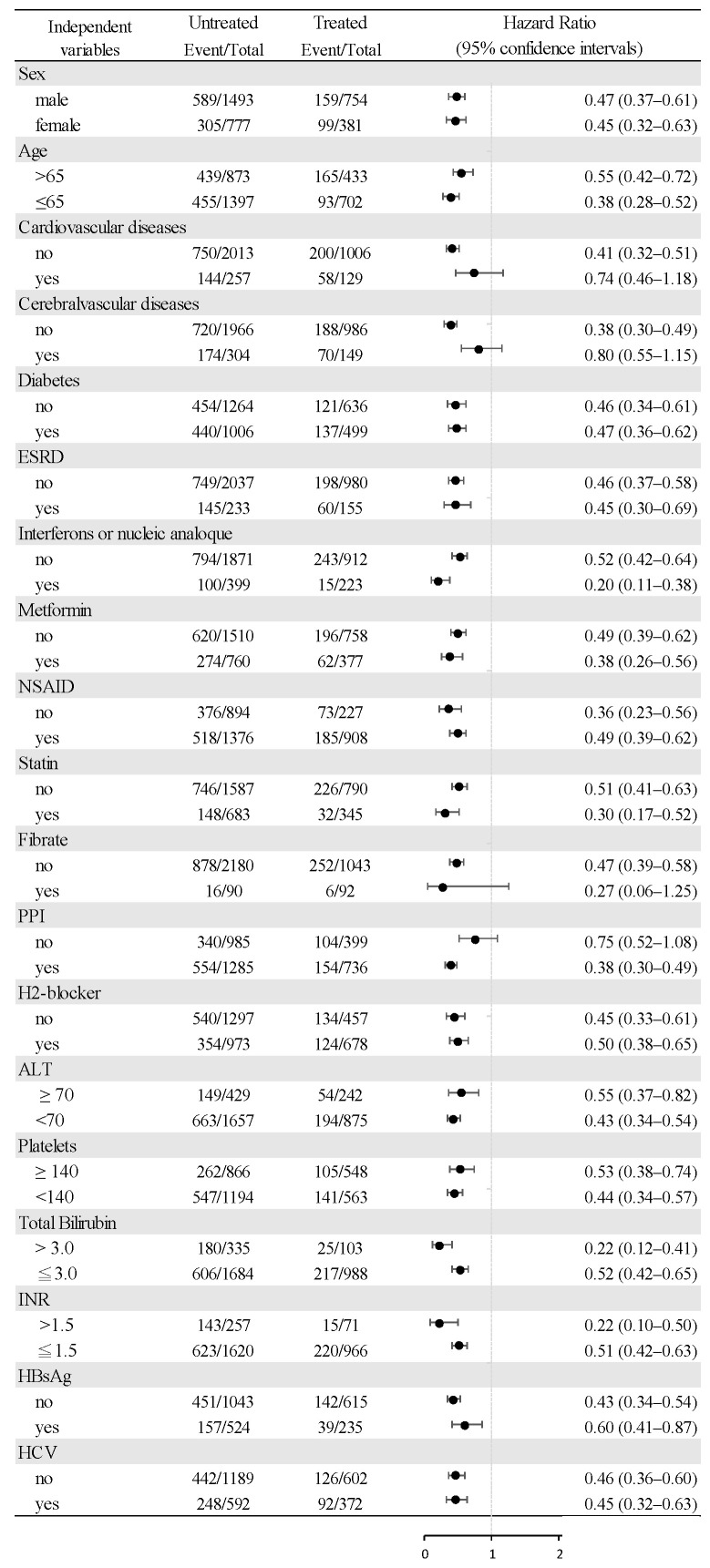
Multivariable stratified analyses of the association between daily aspirin and the risk of hepatocellular carcinoma. Comparison of the risk of hepatocellular carcinoma between the aspirin users (≥84 days) and nonusers. Dark circles: OR; horizontal lines: 95% confidence intervals.

**Table 1 cancers-15-02946-t001:** Effects of daily aspirin or APAs use on the incidence of hepatocellular carcinoma ^a,b^.

Clinical Outcome and Treatment Group	Number with Event/ Total No.	Incidence Rate (Per 100 Person-Year)	Univariate Cox Model				Multivariable Cox Model			
			Crude HR	(95% CI)		*p*-Value	Adjusted HR	(95% CI)		*p*-Value
3-year outcomes										
Incident HCC										
Aspirin	47/1135	1.49	0.46	0.31	0.69	0.0002	0.57	0.37	0.87	0.009
APA ^c^	93/1491	2.49	0.62	0.46	0.83	0.0015	0.82	0.59	1.14	0.243
5-year outcomes										
Incident HCC										
Aspirin	82/1135	2.14	0.55	0.40	0.76	0.0003	0.630	0.45	0.88	0.007
APA ^c^	138/1491	2.90	0.63	0.49	0.82	0.0005	0.81	0.61	1.08	0.151

^a^ Using the Fine and Gray method to take all-cause mortality as a competing risk of incident HCC; ^b^ PSM variables: sex, age, cardiovascular disease, cerebrovascular attack, diabetes mellitus, metformin, and statin; ^c^ APA: non-aspirin antiplatelet agents (Clopidogrel, Dipyridamole, Ticagrelor, Iloprost, and Tirofiban); CI: confidence interval; HR: hazard ratio; PSM: propensity score matching.

**Table 2 cancers-15-02946-t002:** Correlation of the duration of aspirin use with the risk of hepatocellular carcinoma.

Event and Duration of Low-Dose Aspirin	Univariate Cox Model				Multivariable Cox Model			
	Crude HR	(95% CI)		*p*-Value	Adjusted HR	(95% CI)		*p*-Value
Aspirin ^a^								
Nonuser	1.00 (reference)				1.00 (reference)			
3 m to <1 y	0.73	0.52	1.03	0.071	0.88	0.58	1.34	0.553
1 to <3 y	0.73	0.50	1.07	0.103	0.56	0.31	0.99	0.048
≥3 y	0.35	0.21	0.60	0.0001	0.37	0.18	0.76	0.007
Aspirin ^b^								
Nonuser	1.00 (reference)					1.00 (reference)		
3 m to <1 y	0.93	0.62	1.39	0.723	0.85	0.55	1.31	0.455
1 to <3 y	0.50	0.28	0.90	0.020	0.54	0·30	0.98	0.044
≥3 y	0.34	0.17	0.68	0.002	0.39	0.19	0.79	0.009

^a^ PSM variables: sex, age, cardiovascular disease, cerebrovascular attack, diabetes mellitus, metformin, and statins; ^b^ PSM variables: sex, age, cardiovascular disease, cerebrovascular attack, diabetes mellitus, metformin, statin, serum ALT, total bilirubin, HBsAg, anti-HCV antibody, platelet count, and prothrombin time by international normalized ratio.

**Table 3 cancers-15-02946-t003:** Effect of daily aspirin on the risk of gastrointestinal bleeding ^a,b^.

Clinical Outcome and Treatment Group	Multivariable Cox Model			
	Adjusted HR	(95% CI)		*p*-Value
Patients without previous GI bleeding				
3-year outcomes	0.66	0.49	0.90	0.009
5-year outcomes	0.65	0.50	0.86	0.002
Patients with previous GI bleeding				
3-year outcomes	0.51	0.36	0.71	<0.0001
5-year outcome	0.51	0.36	0.72	0.0001

^a^ PSM by sex, age, cardiovascular disease, cerebrovascular attack, diabetes mellitus, metformin, statin, serum ALT, total bilirubin, HBsAg, anti-HCV antibody, platelet count, and prothrombin time by international normalized ratio; ^b^ Using the Fine and Gray method to take all-cause mortality as a competing risk of incident GI bleeding; ALT: alanine aminotransferase; CI: confidence interval, CVD: cardiovascular disease; CVA: cerebrovascular attack; DM: diabetes mellitus; HBV: HBsAg-positive; HCV: anti-HCV-antibody-positive; HR: hazard ratio; INR: international normalized ratio; PSM: propensity score matching.

## Data Availability

The datasets used and analyzed for this study are available from the corresponding author upon request.

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
