# Peer review of "Daily Aspirin Reduced the Incidence of Hepatocellular Carcinoma and Overall Mortality in Patients with Cirrhosis"

_cancers, 2023, doi:10.3390/cancers15112946_

Round 1

Reviewer 1 Report

This manuscript is a meaningful study that showed that the use of aspirin in patients with liver cirrhosis does not increase the bleeding risk and lowers the incidence and mortality of HCC using big data with CGRD. In previous prospective study conducted in patients with chronic viral hepatitis B and chronic viral hepatitis C, it was reported that aspirin use lowered the incidence of HCC, and meta-analysis also showed the similar result. However, these studies have a limited number of cirrhotic cases and remain to be further clarified due to problems such as bleeding risk in liver cirrhosis. Moreover, HCC preventing effects of aspirin were shown to be related to daily aspirin duration. There are few studies to date on the association between aspirin use and HCC risk in patients with liver cirrhosis without subgroup analysis. In recent Hepatology paper (Jang, et al.), the authors demonstrated that cirrhosis had a significant effect on the association between aspirin use and HCC risk, and an association of aspirin use with HCC risk was not evident (aSHR, 1.00; 95% C.I., 0.85-1.18). This may be because, no matter how well designed, it is impossible to completely correct the etiology of liver cirrhosis, smoking, and anti-viral therapy. Therefore, further research is needed based on this study.

There are a few things I would like to point out to improve the quality of the paper.

1. Previous studies have shown that the benefits of aspirin were aspirin dose related. How specifically was the aspirin dose used?

2. The authors performed propensity score matching using PT INR in Study B and D. Patients taking anticoagulation such as warfarin or NOAC, DOAC might have prolongation of PT INR regardless of liver function. How did you solve these problems?

3. The authors conducted a multivariable stratified analysis in Figure 3. Did you analyze patients with and without NAFLD or NASH separately?

4. The subject of this manuscript is that the use of daily aspirin reduces the incidence of HCC and lowers overall mortality in patients with LC. However, most of the authors' discussion focused on GI bleeding with aspirin use (side effect). It would be good to summarize the mechanism related to lowering the HCC risk described in the Introduction, and supplement the related literature in the discussion section.

5. Please describe the limitations of this study.

6. On page 10, lines 301-303, "Nevertheless, co-administration of proton pump inhibitors or histamine receptor inhibitors with aspirin or APA users contributes to no increase in GI bleeding risk in our cirrhotic patients." does not relate to the preceding sentence. Also, the last part of the discussion leads to an abrupt conclusion. Please describe and supplement this part in more detail.

Overall, the English language quality is relatively good, but it is judged that some sentences need improvement.

Author Response

April 28, 2023

Professor Dr. Samuel C. Mok

Editor-in-Chief

Cancers

Dear Professor Dr. Samuel C. Mok

We appreciate the comments from the editor and reviewers. We also thank you for kindly letting us revise our manuscript “Daily Aspirin Reduced the Incidence of Hepatocellular Carcinoma and Overall Mortality in Patients with Cirrhosis.” for publication in your journal, Cancers. We have fully taken the reviewers’ comments to revise the manuscript. 

Please see our response to the reviewers’ comments point-by-point and the locations that highlighted these changes in the revised manuscript.

 Response to Reviewer 1: Comments and Suggestions for Authors

This manuscript is a meaningful study that showed that the use of aspirin in patients with liver cirrhosis does not increase the bleeding risk and lowers the incidence and mortality of HCC using big data with CGRD. In previous prospective study conducted in patients with chronic viral hepatitis B and chronic viral hepatitis C, it was reported that aspirin use lowered the incidence of HCC, and meta-analysis also showed the similar result. However, these studies have a limited number of cirrhotic cases and remain to be further clarified due to problems such as bleeding risk in liver cirrhosis. Moreover, HCC preventing effects of aspirin were shown to be related to daily aspirin duration. There are few studies to date on the association between aspirin use and HCC risk in patients with liver cirrhosis without subgroup analysis. In recent Hepatology paper (Jang, et al.), the authors demonstrated that cirrhosis had a significant effect on the association between aspirin use and HCC risk, and an association of aspirin use with HCC risk was not evident (aSHR, 1.00; 95% C.I., 0.85-1.18). This may be because, no matter how well designed, it is impossible to completely correct the etiology of liver cirrhosis, smoking, and anti-viral therapy. Therefore, further research is needed based on this study.

There are a few things I would like to point out to improve the quality of the paper.

  1. Previous studies have shown that the benefits of aspirin were aspirin dose related. How specifically was the aspirin dose used?

Response 1: Thank you for the comment. Currently, including in our cohort, aspirin is used to prevent platelet aggregation to prevent from vascular occlusion disorders, such as coronary and cerebral ischemia. Daily dose of aspirin is ranged from 75 to 100 mg. In our cohort, the daily aspirin dose in most patients are 100 mg. Herein, we also demonstrate a positive correlation between the daily aspirin duration with the reduction of HCC incidence. (line 109)

  1. The authors performed propensity score matching using PT INR in Study B and D. Patients taking anticoagulation such as warfarin or NOAC, DOAC might have prolongation of PT INR regardless of liver function. How did you solve these problems?

Response 2: Thank you for the comment. Warfarin or NOAC, DOAC analysis are not specifically included in this study. In addition to aspirin, we also analyzed the effects of non-aspirin antiplatelet agents, not anticoagulant agents.

  1. The authors conducted a multivariable stratified analysis in Figure 3. Did you analyze patients with and without NAFLD or NASH separately?

Response 3: Thank you for the comment. We apologize for not analyzing patients with and without NAFLD or NASH separately. This point has been addressed in the limitation paragraph.

  1. The subject of this manuscript is that the use of daily aspirin reduces the incidence of HCC and lowers overall mortality in patients with LC. However, most of the authors' discussion focused on GI bleeding with aspirin use (side effect). It would be good to summarize the mechanism related to lowering the HCC risk described in the Introduction, and supplement the related literature in the discussion section.

Response 4: Thank you for the comment. As suggested, we have summarized the mechanisms by which aspirin might reduce the occurrence of HCC in the Introduction and Discussion.:

In the introduction:

Lines 62-65. “Platelets could be release of multiple factors including TXA2, ADP, angiogenic factors (VEGF, FGF, PDGF) and growth factors (IGF-I, TGF-β1, SDF-1, and direct interaction with leucocytes and endothelial cells, then promote cancer cell proliferation, angiogenesis and metastasis [11].”

Lines 75-77. “Aspirin therapy was associated with only a small increased risk of gastrointestinal bleeding in patients with hepatitis B or C virus, alcohol-related liver disease or nonalcoholic steatohepatitis [11].”

In the discussion: Lines 328-334.

“On the other hand, Ma et al. reported that aspirin use was independently associated with a reduced risk of HCC incidence, recurrence, and death, but an increased risk of bleeding with aspirin use was noted [33]. Regina Zi Hwei Tan et al., studies adults with hepatitis B or C virus, alcohol-related liver disease or nonalcoholic steatohepatitis, but not focus cirrhotic patient, and administered antiplatelet agent (such as aspirin) use reduced HCC incidence, improved liver-related mortality, and with a small increased risk of gastrointestinal bleeding events [11].”

  1. Please describe the limitations of this study.

Response 5: Thank you for the comment. We have addressed the limitations of this study at the end of Discussion (Lines 337-340):

“The limitations of this study include a retrospective cohort study design; only follow-up 5 years, no NAFLD or NASH stratification analysis, and no analyses on the effects of warfarin, NOAC, or DOAC on HCC incidence, overall mortality, and risk of GI bleeding. Large-scale prospective studies are necessary to validation of our observations.”

  1. On page 10, lines 301-303, "Nevertheless, co-administration of proton pump inhibitors or histamine receptor inhibitors with aspirin or APA users contributes to no increase in GI bleeding risk in our cirrhotic patients." does not relate to the preceding sentence. Also, the last part of the discussion leads to an abrupt conclusion. Please describe and supplement this part in more detail.

Response 6: Thank you for the comment. As suggested, we have further discussed what have been reported regarding the risk of daily aspirin for GI bleeding in the Discussiion (Lines 328-334).

On the other hand, Ma et al. reported that aspirin use was independently associated with a reduced risk of HCC incidence, recurrence, and death, but an increased risk of bleeding with aspirin use was noted [33]. Regina Zi Hwei Tan et al., studies adults with hepatitis B or C virus, alcohol-related liver disease or nonalcoholic steatohepatitis, but not focus cirrhotic patient, and administered antiplatelet agent (such as aspirin) use reduced HCC incidence, improved liver-related mortality, and with a small increased risk of gastrointestinal bleeding events [11].

  1. Comments on the Quality of English Language: Overall, the English language quality is relatively good, but it is judged that some sentences need improvement.

Response 7: Thank you for the comment. We will have our manuscript been English edited by a professional English writing editor once it accepted for publication.

Please do not hesitate to contact us if you have any questions regarding the revision of our manuscript. We look forward to hearing from your favorable response soon!

Sincerely yours,

Sen-Yung Hsieh, MD, PhD

Professor

Department of Gastroenterology and Hepatology

Chang Gung Memorial Hospital, Linkou, Taoyuan 333

Taiwan

Siming.shia@msa.hinet.net; siming@cgmh.org.tw

+886-975368031

Reviewer 2 Report

“Multivariate” should be “Multivariable” – there are major difference between the two methods.

What is the definition of “low-dose” aspirin? What are the mean and SD of aspirin dose for the exposed group?

Is aspirin sold over-the-counter? How could the authors know who had exposed to aspirin or not if it sold over-the-counter – since it is not need the prescription?

Since involved propensity score (PS) matching, should also showing the standardized difference (stddiff)between groups on all the variables that generated the PS. A double check to ensure all underline conditions are balanced, stddiff<0.1, between groups.

Line 135, should specify the relative risk of what outcome(s). Death or/and cancer?

Table 1 got cut-off by pages – should avoid it and started in a new page. Hard to read in current format. Meanwhile, suggest results listed as 2 decimal points and p-value 3 decimal points – to avoid too many numbers listed – key-points were not be obvious when too many numbers listed.

Figure 3, since all HRs were less than 2. The x-axis of forest plots should be ranged from 0 to 2 – so the HRs and CI will be larger and clearer. Once again, suggest results listed as 2 decimal points.

Font in the tables were too large while font in the figures were relative too small. Should be consistent font size.

Please replaced “Multivariate” to “Multivariable” in all tables, figures and text.

Will GI bleeding increased when follow-up longer than five years? Worth discussion.

Author Response

April 28, 2023 

Professor Dr. Samuel C. Mok

Editor-in-Chief

Cancers

Dear Professor Dr. Samuel C. Mok

We appreciate the comments from the editor and reviewers. We also thank you for kindly letting us revise our manuscript “Daily Aspirin Reduced the Incidence of Hepatocellular Carcinoma and Overall Mortality in Patients with Cirrhosis.” for publication in your journal, Cancers. We have fully taken the reviewers’ comments to revise the manuscript. 

Please see our response to the reviewers’ comments point-by-point and the locations that highlighted these changes in the revised manuscript.

Response to Reviewer 2 comments

  1. “Multivariate” should be “Multivariable” – there are major difference between the two methods.

Response 1: Thank you for the comment. As suggested, “Multivariate” in Lines 23, 208, 246, and 285 have been changed to “Multivariable” in all tables, figures and text

  1. What is the definition of “low-dose” aspirin? What are the mean and SD of aspirin dose for the exposed group?

Response 2: Thank you for the comment. Aspirin, such as 75 to 100 milligrams (mg) per day, is the definition of low-dose, and usually aspirin 500 mg or over was defined high dose for pain or fever control before. Physicians almost prescribe aspirin 75-100mg per day in recent third decade. High dose aspirin was replaced by acetaminophen or NSAID for pain or fever control in clinical practice. We had no data of the mean and SD of aspirin dose for the exposed group.

  1. Is aspirin sold over-the-counter? How could the authors know who had exposed to aspirin or not if it sold over-the-counter – since it is not need the prescription?

Response 3: Thank you for the comment. Yes! We do not know who had exposed to aspirin or not if it sold over-the-counter, but this bias was minimal effect after dilutional by big database of CGRD. Notably, more than 99% of Taiwanese people are covered by the National Health Insurance Program. Daily aspirin prescribed by physicians is reimbursed by the program.

  1. Since involved propensity score (PS) matching, should also showing the standardized difference (stddiff)between groups on all the variables that generated the PS. A double check to ensure all underline conditions are balanced, stddiff<0.1, between groups.

Response 4: Thank you for the comment. Yes! PSM by sex, age, comorbidities standardized difference (stddiff) <0.1, between groups showed in revised supplement Table S1, then confounding medication and laboratory data was incorporated.

  1. Line 135, should specify the relative risk of what outcome(s). Death or/and cancer?

Response 5: Thank you for the comment. Lines 140-141, the relative risk of HCC and GI bleeding.

  1. Table 1 got cut-off by pages – should avoid it and started in a new page. Hard to read in current format. Meanwhile, suggest results listed as 2 decimal points and p-value 3 decimal points – to avoid too many numbers listed – key-points were not be obvious when too many numbers listed.

Response 6: Thank you for the comment. Table 1 had been started in a new page. Results have been listed as 2 decimal points and p - value 3 decimal points except p-value keep original data if p < 0.001.

  1. Figure 3, since all HRs were less than 2. The x-axis of forest plots should be ranged from 0 to 2 – so the HRs and CI will be larger and clearer. Once again, suggest results listed as 2 decimal points.

Response 7: Thank you for the comment. Lines 231-232. Figure 3 had been revision as suggested.

  1. Font in the tables were too large while font in the figures were relative too small. Should be consistent font size.

Response 8: Thank you for the comment. As suggested, we have made Font and size consistent in the tables and figures.

  1. Please replaced “Multivariate” to “Multivariable” in all tables, figures and text.

Response 9: Thank you for the comment. We have replaced  “Multivariate” to “Multivariable” all the tables, figures, and the text (Lines 23, 208, 246, 285).

  1. Will GI bleeding increased when follow-up longer than five years? Worth discussion.

Response 10: Thank you for the comment. We analyzed GI bleeding for up to 5 years. 

Please do not hesitate to contact us if you have any questions regarding the revision of our manuscript. We look forward to hearing from your favorable response soon!

Sincerely yours,

Sen-Yung Hsieh, MD, PhD

Professor

Department of Gastroenterology and Hepatology

Chang Gung Memorial Hospital, Linkou, Taoyuan 333

Taiwan

Siming.shia@msa.hinet.net; siming@cgmh.org.tw

+886-975368031

Reviewer 3 Report

Dear Authors,

Thank you for the paper. It is an ongoing issue, so of interest. However, I have some questions about the methodology.

·      Line 89: why did you set three times as a limit?

·      Line 102: why is 20 years old a limit?

·      Line 105: why 84 days?

·      Line 113: on what evidence you based liver fibrosis degree on platelet count?

·      To me, it's not clear how you defined a patient with cirrhosis. How many are diagnosed with biopsy? How many only clinically? If the patient is cirrhotic, why you decided to further divide the population into two arms with different degrees of cirrhosis?

·      What is the CGRD? Who keeps the record, the statistician or the physician? What is the accuracy of the database?

Furthermore, a recent paper has been published about HCC/aspirin (https://doi.org/10.1007/s00228-022-03414-y). What are the differences and strengths of your paper compared to this last publication? Other papers already evaluated the role of aspirin on HCC (Tan RZH, Lockart I, Abdel Shaheed C, et al. Systematic review with meta-analysis: The effects of non-steroidal anti-inflammatory drugs and anti-platelet therapy on the incidence and recurrence of hepatocellular carcinoma), please add to the discussion and comment on that.

The English needs minor revision. 

Author Response

April 28, 2023

Professor Dr. Samuel C. Mok

Editor-in-Chief

Cancers 

Dear Professor Dr. Samuel C. Mok

We appreciate the comments from the editor and reviewers. We also thank you for kindly letting us revise our manuscript “Daily Aspirin Reduced the Incidence of Hepatocellular Carcinoma and Overall Mortality in Patients with Cirrhosis.” for publication in your journal, Cancers. We have fully taken the reviewers’ comments to revise the manuscript. 

Please see our response to the reviewers’ comments point-by-point and the locations that highlighted these changes in the revised manuscript.

Response to Reviewer 3 comments

Dear Authors,

Thank you for the paper. It is an ongoing issue, so of interest. However, I have some questions about the methodology.

  1. Line 89: why did you set three times as a limit?

Response 1: Thank you for the comment. We set “at least” three times as a limit for re-confirm disease diagnosed.

  1. Line 102: why is 20 years old a limit?

Response 2: Thank you for the comment. Adult was defined > 20 y/o in Taiwan.

  1. Line 105: why 84 days?

Response 3: Thank you for the comment. Treatment defined as the duration of at least 84 days was based on the Taiwan National Health Insurance Program reimbursement policy, in which 28 days, instead of 30 days, were counted as one month.

  1. Line 113: on what evidence you based liver fibrosis degree on platelet count?

Response 4: Thank you for the comment. Line 119 platelet counts (liver fibrosis degree) [23].

Zhong, L. K., G. Zhang, S. Y. Luo, W. Yin and H. Y. Song. "The value of platelet count in evaluating the degree of liver fibrosis in patients with chronic hepatitis b." J Clin Lab Anal 34 (2020): e23270. 10.1002/jcla.23270. https://www.ncbi.nlm.nih.gov/pmc/articles/PMC7370727/pdf/JCLA-34-e23270.pdf.

  1. To me, it's not clear how you defined a patient with cirrhosis. How many are diagnosed with biopsy? How many only clinically? If the patient is cirrhotic, why you decided to further divide the population into two arms with different degrees of cirrhosis?

Response 5: Thank you for the comment. The diagnosis of cirrhosis was based on imaging studies, including sonography and computer tomography, and clinical manifestations of portal hypertension (including esophageal and/or gastric varices, ascites, and hypersplenism thrombocytopenia) or insufficiency of hepatic functions (including hypoalbuminemia, prolonged prothrombin time, and hepatic encephalopathy). Decompensated cirrhosis indicate patients presenting with at least two of the above symptoms, which significant impacts patients’ outcomes.

  1. What is the CGRD? Who keeps the record, the statistician or the physician? What is the accuracy of the database?

Response 6: Thank you for the comment. “The Chang Gung Research Database (CGRD), a regularly updated and well-validated tool, offered comprehensive diagnostic, laboratory, medication, and outpatient and hospitalization information of patients followed long-term at six leading hospitals in different regions of Taiwan.” These statements have been presented in lines 80-83.

  1. Furthermore, a recent paper has been published about HCC/aspirin (https://doi.org/10.1007/s00228-022-03414-y). What are the differences and strengths of your paper compared to this last publication? Other papers already evaluated the role of aspirin on HCC (Tan RZH, Lockart I, Abdel Shaheed C, et al. Systematic review with meta-analysis: The effects of non-steroidal anti-inflammatory drugs and anti-platelet therapy on the incidence and recurrence of hepatocellular carcinoma), please add to the discussion and comment on that.

Response 7: Thank you for the comment. As suggested we have added the related references in the Introduction and Discussion:

In the introduction, lines 62-65:

Platelets could be release of multiple factors including TXA2, ADP, angiogenic factors (VEGF, FGF, PDGF) and growth factors (IGF-I, TGF-β1, SDF-1, and direct interaction with leucocytes and endothelial cells, then promote cancer cell proliferation, angiogenesis and metastasis [11].

Lines 75-77: Aspirin therapy was associated with only a small increased risk of gastrointestinal bleeding in patients with hepatitis B or C virus, alcohol-related liver disease or nonalcoholic steatohepatitis [11].

In the discussion, lines 328-334:

On the other hand, Ma et al. reported that aspirin use was independently associated with a reduced risk of HCC incidence, recurrence, and death, but an increased risk of bleeding with aspirin use was noted [33]. Regina Zi Hwei Tan et al., studies adults with hepatitis B or C virus, alcohol-related liver disease or nonalcoholic steatohepatitis, but not focus cirrhotic patient, and administered antiplatelet agent (such as aspirin) use reduced HCC incidence, improved liver-related mortality, and with a small increased risk of gastrointestinal bleeding events [11].

  1. Comments on the Quality of English Language: The English needs minor revision. 

Response 8: Thank you for the comment. Draft will be sent for professional English editing once it accepted for publication.

Please do not hesitate to contact us if you have any questions regarding the revision of our manuscript. We look forward to hearing from your favorable response soon!

Sincerely yours,

Sen-Yung Hsieh, MD, PhD

Professor

Department of Gastroenterology and Hepatology

Chang Gung Memorial Hospital, Linkou, Taoyuan 333

Taiwan

Siming.shia@msa.hinet.net; siming@cgmh.org.tw

+886-975368031

Round 2

Reviewer 1 Report

- The authors are judged to have faithfully answered the points pointed out by the reviewers. There are no further comments.

Author Response

Response to Reviewer 1 Comments

The authors are judged to have faithfully answered the points pointed out by the reviewers. There are no further comments.

Response: Thank you for the comment.

Reviewer 3 Report

Thank you for the replies. Despite the modification, I would expect a more detailed discussion. Also, the study is of interest but there are currently many other papers on this topic, and I don't see the innovation that this paper can add in a journal such as Cancers. 

No major issue

Author Response

Response to Reviewer 3 comments

Quality of English Language
(x) Minor editing of English language required

Thank you for the replies. Despite the modification, I would expect a more detailed discussion. Also, the study is of interest but there are currently many other papers on this topic, and I don't see the innovation that this paper can add in a journal such as Cancers. 

Response: Thank you for the comment. As suggested, we have made a more detailed discussion about the current status of anti-platelet agents in preventing liver cancers. Please refer to the Discussion, Lines 334-338. “P2Y12 antagonists or reagents directly blocking platelet-derived GPIbα or anti-platelet therapy related pathways to block NASH to HCC transition [13]. Previous reports have shown that anti-platelet therapy significantly reduced the risk of HCC after propensity score-matched [34]. APA had a trend to reduce HCC incidence and less than aspirin efficacy in our cirrhotic patients.”

Comments on the Quality of English Language

Response: Thanks for the comment. Although this manuscript has been professionally edited in English before this submission, we will be pleased to have your MPDI English editor team polish the English writing of this manuscript further once it has been accepted for publication in your journal, Cancers.

Round 3

Reviewer 3 Report

Ok for publication

No major issue